**Data Availability Statement:** All relevant data are within the paper and its Supporting Information files.

# Medication adherence and its impact on glycemic control in type 2 diabetes mellitus patients with comorbidity: A multicenter cross-sectional study in Northwest Ethiopia

**Ashenafi Kibret Sendekie**[ID][1]*, **Adeladlew Kassie Netere**[ID][1], **Asmamaw Emagn Kasahun**[2], **Eyayaw Ashete Belachew**[ID][1]

**1** Department of Clinical Pharmacy, School of Pharmacy, College of Medicine and Health Science, University of Gondar, Gondar, Ethiopia, **2** Department of Pharmaceutics, School of Pharmacy, College of Medicine and Health Science, University of Gondar, Gondar, Ethiopia

* ashukib02@yahoo.com, Ashenafi.Kibret@uog.edu.et

## Abstract

### Background

Medication nonadherence in patients with chronic diseases, particularly in type 2 diabetes mellitus (T2DM) with comorbidity, has continued to be the cause of treatment failure. The current study assessed medication adherence and its impact on glycemic control in T2DM patients with comorbidity.

### Methods

An institutional-based multicenter cross-sectional study was conducted among T2DM patients with comorbidity at the selected hospitals in Northwest Ethiopia. Medication adherence was measured using a structured questionnaire of the General Medication Adherence Scale (GMAS). A logistic regression model was used to identify predictors of the level of medication adherence and glycemic control. P < 0.05 at 95% confidence interval (CI) was statistically significant.

### Results

A total of 403 samples were included in the final study. This study showed that more than three-fourths (76.9%) of the participants were under a low level of medication adherence. Source of medication cost coverage [AOR = 10.593, 95% CI (2.628–41.835; P = 0.003], monthly income (P < 0.00), self-monitoring of blood glucose (SMBG) practice [AOR = 0.266, 95% CI (0.117–0.604); P = 0.002], number of medications [AOR = 0.068, 95% CI (0.004–0.813); P = 0.014] and medical conditions [AOR = 0.307, 95% CI (0.026–0.437); P = 0.018] were found to be significant predictors of medication adherence. Significantly, majority (74.7%) of participants had poor levels of glycemic control. Patients who had a high level of medication adherence [AOR = 0.003, 95% CI (0.000–0.113); P = 0.002] were found less likely to have poor glycemic control compared with patients who were low adherent to their medications.

**Funding:** The author(s) received no specific funding for this work.

**Competing interests:** The authors have declared that no competing interests exist.

**Abbreviations:** ADA, American diabetes association; BMI, body mass index; FBG, fasting blood glucose; HbA1c, glycosylated hemoglobin; NPH, Neutral Protamine Hagedorn; OADs, Oral Antidiabetics; T2DM, Type 2 Diabetes Mellitus.

## Conclusion

The current study concluded that medication adherence was low and significantly associated with poor glycemic control. Number of medical conditions and medications were found to be associated with medication adherence. Management interventions of T2DM patients with comorbidity should focus on the improvement of medication adherence.

## Introduction

Diabetes mellitus (DM) continues to be a public health problem worldwide with the number of people presenting with diabetes estimated to be 783 million by 2045 [1]. The prevalence of T2DM in developing countries has increased rapidly worldwide and accounts for more than 95% of diabetes cases [2]. The majority (three-fourths) of the diabetes patients are living in low and middle-income countries [3]. In Africa, it was reported to be 24 million in 2021 and estimated to reach 55 million (5%) by 2045 [1]. This makes diabetes becomes among the most common public health threats. The growth rate of DM has also increased in Ethiopia, and there is an observable change in lifestyle and significant increases in population and urbanization, which are the identified risk factors for DM. About more than two and a half million adults in Ethiopia have currently live with diabetes [4], and the prevalence has increased dramatically from 3.8% to 5.2% [5]. These make Ethiopia as one of the sub-Saharan Africa countries with the largest population of diabetes. While T2DM is estimated to be higher than this figure and the pooled prevalence reaches 8% [6].

The main treatment goal of diabetes patients is to maintain glycemic control and prevent diabetes-related complications, and morbidities and mortality [7]. However, suboptimal management of patients leads to treatment failure and complications [8]. For treating patients with diabetes, self-monitoring of blood glucose (SMBG), lifestyle modifications and the administration of medications are the recommended management interventions [9]. Nevertheless, medication non-adherence to the prescribed regimens has been continued to be a barrier of effective treatment outcome in the management of chronic disease conditions [7, 10]. Non-adherence to prescribed medication regimes contributes to treatment failure, risk of hospitalization, and morbidity and mortality in patients with long-term medication therapy [11].

Globally, a significant proportion of T2DM patients are non-adherent to their prescribed medications. Even, in the developed states around 50% of patients are non-adherent to their long-term medication and it is also much higher in low-and middle-income countries [12]. Personal and socio-demographic characteristics as well as medication regimen complexity, clinical characteristics and the number of medical conditions are factors that influence medication adherence in patients with chronic diseases [13–17]. Different studies have shown that non-adherence to prescribed medications in patients with diabetes mellitus is reported to be high and ranges from 6.3% to 87% [7, 18–20]. Evidence suggests that non-adherence to diabetes medications affects glycemic control that leads to complications associated with diabetes progression, hospitalizations, morbidity and mortality [3, 17, 21–24]. This in turn increases the risk of negative consequences and high medical costs with considerable direct and indirect problems to the sustainability of the healthcare system [3, 17, 21–24].

Studies also revealed that knowledge about diabetes and medications, level of patient educational status, occupational status, duration of diabetes and its treatments are among the factors that contribute to medication adherence [25–28]. Majority of patients with T2DM in Ethiopia are with comorbidities such as hypertension, dyslipidemia and macrovascular complications and had significantly poor glycemic control [28, 29]. Polypharmacy and medication regimen

complexity have been considered to be the most factors of poor adherence to medications among patients with chronic disease conditions and comorbidities [30, 31]. The burden of diabetes has increased and the prevalence of comorbidities is much higher among T2DM patients in Ethiopia. However, there is a paucity of real-world evidence, particularly in the study settings, which assessed medication adherence and its impact on glycemic control among T2DM patients with comorbidities. Therefore, this study assessed medication adherence among patents with T2DM and comorbidity at selected hospitals in Northwest Ethiopia. Moreover, the study also assessed the impact of medication adherence on glycemic control in patients with T2DM.

## Methods and materials

### Study design, settings and participants

An institutional-based multicenter cross-sectional study was conducted among T2DM patients with comorbidity. The study was employed in outpatient follow-up clinics at selected hospitals in Northwest Ethiopia from January to March 2022. The study area Northwest Ethiopia is the geographical location of the Northwestern part of the Amhara regional state, which is a metropolitan area and one of the Ethiopian government administration regional states. The study samples were recruited from the University of Gondar Comprehensive specialized hospital (UoGCSH), Tibebe-Ghion Comprehensive Specialized hospital (TGCSH), Felege-Hiwot Comprehensive Specialized hospital (FHCSH) and Debre-Tabor Comprehensive Specialized hospital (DTCH). The study hospitals are governmental hospitals, which have been served more than 20 million population and were randomly selected among several public and university hospitals found in the region. All the selected hospitals have chronic follow-up clinics, including diabetes patient care.

To be included in the study, participants should be adults (aged 18 years or older), diagnosed with T2DM, and are diagnosed with at least one comorbidity. In addition, they have been on treatment for at least a minimum of three months. While patients who were unable to communicate because of neurological or psychiatric illness, and/or severely ill patients, pregnant mothers, patients with incomplete medical records were excluded from this study.

### Sample size determination and sampling techniques

We determined the sample size using a single population proportion formula by considering; response distribution, P = 0.5 (50%), and at 95% confidence interval, the marginal error was 5% for the two-tailed type-I error (Zα = 1.96). The sample size was to be 385. Finally, considering a 10% potential nonresponse to the interview and/or missed or lost data on the patient's medical record, 423 patients were approached in the final study. Then, the final sample size was proportionally divided into the selected hospitals to take a representative sample from each hospital. The number of patients with T2DM in each hospital was taken from records of the previous three follow-up months in the settings. All T2DM patients who fulfilled the inclusion criteria and come for follow-up during the data collection periods were approached until the required sample was achieved. Eventually, proportional to the number of T2DM patients in the selected hospitals; 174, 125, 68, and 56 eligible patients were included at the diabetes follow-up clinics of UoGCSH, FHCSH, DTCSH and TGCSH hospitals, respectively. Study participants from all selected hospitals were included using consecutive sampling technique.

### Data collection instruments and procedures

Data was collected using a structured questionnaire. The data collection tool was prepared in English version after reviewing different related literature on similar topics and some

modifications were made considering the local clinical settings. It was translated to local language, Amharic for making easy for data collection process. The tool was organized with different sections. The first section consisted of socio-demographic sections that included age, sex, weight, BMI, residency, educational status, employment status, physical activity, SMBG practice and cigarette smoking habit of the participants. The second section describes the clinical characteristics of the participants. This section is consisted of a type of medical condition such as comorbidities and complications, number of medical conditions, laboratory tests, blood glucose and blood pressure values and prescribed medications used for treating the study participants. Questionnaires assessing medication adherence is the last section of the data collection instrument.

The data was collected by four experienced nurses and two pharmacists from the hospitals after getting of training on the purpose of the study, data collection instruments and producers and about ethical aspects. The data collectors were engaged voluntarily. After the medical record identification numbers were entered into the Microsoft excel 2013 and checked for repetition, the data were extracted, and the patients were interviewed. Data were collected on direct patient interviews for primary data, and laboratory tests, medical conditions and dosage of medications were recorded from patients' medical records. Laboratory test results were also checked from printed laboratory records.

Treatment intensification and titrations were made according to ADA recommendations. Metformin alone or with insulin and/or glibenclamide were the medication regimens used to treat T2DM in the study settings. The glycemic level of the participants was determined by an average of three different records of FBG, at least one month apart, this was because of inconsistent records of HbA1c in the study settings and included participants. In the resource-limited settings, a very limited number of patients were monitored using HbA1c in a regular fashion. The weight and height of the participants were measured using a digital weight scale and stadiometer as physical examination part.

### Adherence

It indicates the active, voluntary, and collaborative decisional involvement of the patient in a mutually acceptable course of behavior to produce a therapeutic result.

### Body mass index (BMI)

It is measured from weight in kilograms (kg) divided by the square of the patient's height in meters (kg/m2). Based on the world health organization; BMI was classified and interpreted as < 18.5 kg/m2 (underweight), 18.5–24.9 kg/m2 (normal weight), 25–29.5 kg/m2 (overweight) and $\geq$ 30 kg/m2 (obesity).

## Outcome measurements

### Adherence measurement

Medication adherence was measured by using the General Medication Adherence Scale (GMAS), which has an 11-item questionnaire that provides a convenient way of tracking compliance using a combination of subjective and objective measures. Each item had four Likert scores, with a minimum score of 0 and a maximum score of 3. The items are subdivided in to (I) patient behavior-related medication adherence questionnaires (5 items) (II) pill/injection burden due to additional disease related questionnaires (4 items), and (III) the third subsection is payment-related questionnaires (2 items). The GMAS instrument of medication adherence has been used and validated in several studies of different chronic diseases [32–35]. The

English version of the questionnaire is also validated [34] with an internal consistency of the items for its reliability test of Cronbach alpha resulted 0.84 and the item-level content validity indexes were $\geq 0.79$.

The final outcome score used to categorize the medication adherence level as low adherence and high adherence was determined by computing the sum of each item scores. If the overall GMAS score $\leq 26$, the patient was categorized under low adherence and patients were categorized under high adherence if the GMAS score was greater than or equal to 27 out of 33 overall maximum points [36].

### Glycemic control measurement

In this study, the level of glycemic control is measured based on ADA recommendations. Glycemic level in the range of FBG $< 70$ mg/dl and $> 130$ mg/dl to be poor glycemic and FBG of 70–130 mg/dl was good glycemic control. The level of FBG used to determine glycemic control was taken from the average of three recorded FBG's which were measured for at least a month apart.

### Data quality control and statistical analysis

Before the actual data collection, the questionnaire was pre-tested on 5% of the study subjects in one of the study areas (excluded from the final analyses) to ensure completeness and consistency of the data collection tool. Then, an appropriate amendment was employed. The data was collected by experienced nurses and pharmacists after getting training for two days. The supervisor explicitly clarified the purpose of the study and about data collection tools and techniques. The data collection procedure was monitored closely. After the medical record identification numbers were entered into the Microsoft excel 2013 and checked for repetition, the patients were interviewed and simultaneously the data was extracted.

Once the data was collected; quality, completeness, consistency and clarity were checked before any further analysis was performed. Then, it was entered into Epi-Info version 8, and transported and analyzed with the Statistical Packages for Social Sciences (SPSS) version-26. Shapiro-Wilk tests, Q-Q plots, and histograms were used to examine the normal distribution of the data. Categorical variables are presented as frequencies and percentages. While means with standard divisions (±SD) were used to display results for continuous variables. A logistic regression model was used to assess the association of medication adherence and glycemic control, and with other predictor variables. Variables with $p \leq 0.2$ in the bivariate analysis were considered for further analyses in the multivariable analysis to identify predictor variables with medication adherence and glycemic control status. $P < 0.05$ at 95% CI was statistically significant.

### Ethical considerations

Initially, the study was ethically approved by the ethical review committee of the University of Gondar with a reference number of Sop/037/2021. Then, permission confirmation was gained from the selected hospitals to proceed with the study. Participants were asked with both written and verbal consent forms, and after the objectives of the study were briefed, consent was accessed to interview them. Confidentiality was kept and sufficiently anonymized and the study was conducted according to the Helsinki legislation.

## Results

### Socio-demographic and clinical characteristics of the study participants

Out of 423 approached participants, 403 samples were included in the final study. Greater than half (54.8%) of the participants were males with a mean (±SD) age of 55±10.8 years. In

addition to T2DM, most of the participants were comorbid with hypertension (71.2%) followed by dyslipidemia (42.4%). An average of 2.8 (ranges: 2–6) medical conditions per patient were recorded. The average FBG level of the participants was estimated to be 176.0 mg/dl (**Table 1**).

## Medications used for treating participants

A greater proportion of the participants (32.5%) were treated with a combination of metformin plus insulin, and NPH insulin accounts higher proportion 46.9%) from types of insulin regimens. Enalapril (24.3%) and atorvastatin (35.5%) were also commonly prescribed antihypertensive and lipid-lowering agents, respectively. An average of 4.2 (range: 2–9) medications were prescribed per patient. The average daily dose of insulin, metformin and glibenclamide were 17.2 units (range: 10–40), 1356.8 (range: 500–2000) mg and 13.2 (ranges: 5–20) mg, respectively (**Table 2**).

## Level of medication adherence of the study participants

A higher proportion of the participants who responded to the GMAS measuring items that they were missed their medications either mostly or sometimes. Overall, the current findings showed that medication adherence is significantly lower. More than three-fourths of the participants (76.9%) 95% CI (72.7–81.1) were low adherent to their medications, with an average overall GMAS score of 22.08 (ranges:15–33) out of 33 points (**Table 3** and **Fig 1**).

## Determinants of medication adherence

Predictor variables of the level of medication adherence were identified using logistic regression analysis. The multivariable logistic regression model showed that sources of medication cost coverage, monthly income, SMBG practice, number of medications and medical conditions were found to have a significant association with the level of medication adherence. Participants who covered their medication costs out of pocket were found more likely to be low adherent to their medication compared to those who received medications without payment [AOR = 10.593, 95% CI (2.628–41.835); p = 0.003]. Similarly, patients with lower monthly income (< 1500, 1500–2999, and 3000–4999) were also found more likely to have low adherence to their medications compared to patients who had 5000 and higher monthly income [AOR = 13.896, 95% CI (2.598–46.199), AOR = 9.369, 95% CI (2.940–25.785) and AOR = 5.095, 95% CI (2.549–13.308); p < 0.001], respectively. In contrast, patients who could practice SMBG, patients with a lower number of medications (≤ 3) and patients with two medical conditions were found less likely to be low adherent to their medications compared to patients who did not practice SMBG, patients with greater than or equal to six numbers of medications and patients with greater than or equal to five medical conditions: [AOR = 0.266, 95% CI (0.117–0.604); p = 0.002], [AOR = 0.068, 95% CI (0.004–0.813); p = 0.014] and [AOR = 0.307, 95% CI (0.026–0.437); p = 0.018], respectively (**Table 4**).

## Level of glycemic control and its association with medication adherence and other variables

Overall, the average blood glucose level of the participants was far higher than the target level, with an average FBG of 176.0±51.4 mg/dl (ranges: 89–349). Compared to adherent patients (Mn = 130.1) nonadherent participants had significantly worse FBG levels (Mn = 190.9). In terms of the level of glycemic control, around three-fourths (74.7%) of the study participants

**Table 1. Socio-demographic and clinical characteristics of T2DM patients with comorbidity at hospitals in Northwest Ethiopia from January to March, 2022 (N = 403).**

| Socio-demographic variables | | Frequency (%) | Mean (±SD) |
|---|---|---|---|
| Sex | Male | 221 (54.8) | |
| | Female | 182(45.2) | |
| Age in years | | - | 55(±10.8) |
| Weight in Kg. | | - | 65.6(±8.3) |
| Residence | Urban | 237(58.8) | |
| | Rural | 166(41.2) | |
| Educational status | Unable to read and write | 55(13.6) | |
| | Primary school | 133(33) | |
| | Secondary school | 150(37.2) | |
| | University or college and above | 65(16.1) | |
| Occupation | Farmer | 74(18.4) | |
| | Government employee | 103(25.6) | |
| | Self-employed | 98(24.3) | |
| | Student | 43(10.7) | |
| | Unemployed | 63(15.6) | |
| | Others | 22(5.5) | |
| Monthly income (ETH.Birr) | | | 3775.4(±1627.2) |
| Source of medication cost coverage | Health insurance | 233(57.8) | |
| | Out of pocket | 122(30.3) | |
| | Free | 48(11.9) | |
| Body mass index (Kg/M$^2$) | Low | 34 (8.4) | 24.6(±11.2) |
| | Normal | 235 (58.3) | |
| | Over weight | 56 (13.9) | |
| | Obese | 78 (19.4) | |
| Duration since T2DM diagnosis (years) | 1–5 | 30(7.4) | 13.4(±7.8) |
| | 6–10 | 141(35) | |
| | 11–20 | 187(46.4) | |
| | > 20 | 45(11.2) | |
| Cigarette Smoking status | Currently smoker | 69(17.1) | |
| | Previously smoker | 97(24.1) | |
| | Non-smoker at all | 237(58.8) | |
| Alcohol drinking habit | No | 182(45.2) | |
| | Yes | 221(54.8) | |
| Self-monitoring of blood glucose | Yes | 125(31) | |
| | No | 278(69) | |
| Family history of T2DM | Yes | 263(65.3) | |
| | No | 140(34.7) | |
| Physical activity | Sedentary | 181(44.9) | |
| | Moderate | 138(34.2) | |
| | Vigorous | 84(20.8) | |

(*Continued*)

**Table 1.** (Continued)

| Socio-demographic variables | | Frequency (%) | Mean (±SD) |
|---|---|---|---|
| Medical conditions (comorbidities and complications) | Hypertension | 287(71.2) | |
| | Dyslipidemia | 184 (45.7) | |
| | Macrovascular complications | 71 (17.6) | |
| | Hypoglycemia in recent time | 52 (12.9) | |
| | Microvascular complications | 30 (7.4) | |
| | Renal disorders | 22 (5.5) | |
| | Diabetes ketoacidosis | 21(5.2) | |
| | Retroviral infection | 11 (2.7) | |
| | Others* | 24(6) | |
| Number of medical conditions | | - | 2.8(±0.8) |
| **Laboratory parameters** | | | |
| Fasting blood glucose (mg/dl) level | | | 176.0(±51.4) |
| Systolic blood pressure (mmHG) | | | 137.3(±11.6) |
| Diastolic blood pressure (mmHG) | | | 81.3(±9.5) |
| Serum creatinine level (mg/dl) | | | 1.9(±9.2) |
| Total cholesterol level | | | 196(±49.6) |
| Total glyceride level | | | 168.6(±45.6) |

Others*; Bacterial infections, thyrotoxicosis, bronchial asthma, malaria, skin disorders.

had a poor level of glycemic control, and only one-fourth (25.3%) had achieved a target glycemic level.

The multivariable logistic regression model showed that SMBG practice of the patients, level of BMI (Kg/m$^2$) and level of medication adherence were found to have a significant association with the level of glycemic control in patients with T2DM with comorbidity. With holding other variables constant, patients who could practice SMBG [AOR = 0.319, 95% CI (0.056–0.829): p = 0.020], patients who had a normal level of BMI [AOR = 0.280 95% CI (0.002–0.474); p = 0.013], and patients with high medication adherence [AOR = 0.003, 95% CI (0.000–0.113); p = 0.002] were found less likely to have poor glycemic control compared with patients who were not practiced SMBG, patients with obesity and patients who had low medication adherence, respectively (**Table 5**).

## Discussion

This institutional-based multicenter study has gone through highlighting the level of medication adherence using a structured questionnaire of GMAS for chronic diseases and its impact on glycemic control in T2DM patients with comorbidity. Ensuring medication adherence in patients with chronic conditions, especially in multimorbid patients is continued to be the most challenging in healthcare practice because of medication complexity and its multiple burden. The problem is more severe in low-income countries and poor settings where there is a low level of patients' educational status, knowledge about diabetes and medications [25–28]. In Ethiopia, a significant proportion of patients with T2DM have comorbid conditions like hypertension, dyslipidemia and macrovascular and microvascular complications [29]. However, medication adherence can be influenced by the medication regimen complexity and the polypharmacy [30, 31, 37] used to treat these comorbidities. Poor glycemic control because of poor medication adherence can increase the risk of negative consequences and medical costs

**Table 2. Distribution of medications used to the treatment of T2DM patients with comorbidity.**

| Medications | | Frequency (%) | Mean (±SD) |
|---|---|---|---|
| Antidiabetic medications | Metformin plus insulin | 131(32.5) | |
| | Metformin plus glibenclamide | 76(18.9) | |
| | Metformin | 74(18.4) | |
| | Metformin plus glibenclamide plus insulin | 63(15.6) | |
| | Insulin | 59(14.6) | |
| Types of insulin regimens | NPH | 189(46.9) | |
| | Premixed insulin | 68(16.9) | |
| Antihypertensive and cardiovascular agents | Enalapril | 98(24.3) | |
| | Amlodipine | 66(16.4) | |
| | Hydrochlorothiazide | 56(13.9) | |
| | Atenolol | 19 (4.7) | |
| | Metoprolol | 15 (3.7) | |
| | Nifedipine | 12(3) | |
| | Furosemide | 7 (1.7) | |
| Lipid-lowering agents | Atorvastatin | 143(35.5) | |
| | Simvastatin | 48(11.9) | |
| Aspirin (ASA) | | 67(16.6) | |
| Amitriptyline | | 23 (5.7) | |
| TDF/3TC/DTG | | 11(2.7) | |
| Warfarin | | 6 (1.5) | |
| Propyl thiouracil | | 5 (1.2) | |
| Salbutamol plus beclomethasone | | 5 (1.2) | |
| Others* | | 19(1.6) | |
| Number of medications | | | 4.2(±1.4) |
| Average daily dose of insulin (Unit) | | | 17.2(±5.9) |
| Average daily dose of metformin (mg) | | | 1356.8(±1428.9) |
| Average daily dose of glibenclamide (mg) | | | 13.2(±5.1) |
| Average daily dose of Atorvastatin (mg) | | | 43.2(±30.8) |
| Average daily dose of Simvastatin (mg) | | | 26.1(±28.1) |

TDF, Tenofovir disoproxil fumarate; 3TC, Lamivudine; DTG, Dolutegravir; others

* include antibiotics, gastrointestinal drugs and antipains.

with significant impactful problems to the sustainability of the healthcare system [3, 17, 22–24].

The current study showed that a higher proportion of T2DM patients with comorbidity were low adherent to their medications and were found to have a poor level of glycemic control. Participants who covered their medication costs out of pocket and those patients with lower monthly income were found more likely to have low adherence to their medications. However, patients who could practice SMBG, patients with a lower number of medications and those patients with a lower number of medical conditions were found less likely to become low adherent to their medications. Further, this study disclosed that the level of glycemic control was found to have a significant association with the level of medication adherence.

In this study, most of the participants were found to have a low level of medication adherence. This finding is consistent with an earlier study [38], but it is much higher than the other studies [14, 18, 36, 39–41]. The study findings indicate that a significant proportion of patients with comorbidity fail to achieve the expected adherence level of medications. This might be

**Table 3. Medication adherence with respect to GMAS measuring items.**

| | GMAS measuring item descriptions | Adherence response levels n (%) | | | | Mean (±SD) Score |
|---|---|---|---|---|---|---|
| | | Always | Mostly | Some times | Never | |
| 1. | Difficulty in remember to take medications | - | 8[19] | 258(64) | 137(34) | 2.32(±0.5) |
| 2. | Forgetting medications due to busy schedules, travel and other events | - | 30(7.4) | 253(62.8) | 120 (29.8) | 2.22(±0.6) |
| 3. | Discontinuing medications when feeling well | - | 109(27) | 192(47.6) | 102 (25.3) | 1.98(±0.7) |
| 4. | Stopping taking medications due to adverse effects | - | 4(1) | 338(83.9) | 61(15.1) | 2.14(±0.4) |
| 5. | Stop medications without telling a doctor | - | 49(12.2) | 240(59.6) | 114 (28.3) | 2.16(±0.6) |
| 6. | Discontinuing medications due to other medicines for additional diseases | 1(0.2) | 81(20.1) | 254(63) | 67(16.6) | 1.96(0.6) |
| 7. | Find it hassle to remember medications due to medication regimen complexity | 2(0.5) | 61(15.1) | 279(69.2) | 61(15.1) | 1.99(±0.6) |
| 8. | Missing medicines due to progression of disease and addition of new medicines in the last month | - | 85(21.1) | 245(60.8) | 73(18.1) | 1.97(±0.6) |
| 9. | Altering medication regimen, dose and frequency | - | 123 (30.5) | 214(53.1) | 66(16.4) | 1.86(±0.7) |
| 10. | Discontinuing medications because they are not worth for the money | 1(0.2) | 124 (30.8) | 251(62.3) | 27(6.7) | 1.75(±0.6) |
| 11. | Find it difficult to buy medicines because they are expensive | 1(0.2) | 143 (35.5) | 226(56.1) | 33(8.2) | 1.72(±0.6) |
| **Overall GMAS mean score** | | | | | | **22.08(4.4)** |

Note: Always = 0; mostly = 1; sometimes = 2; never = 3.

because patients may discontinue medications due to other medications for additional diseases and or it might be difficult to remember medications due to medication regimen complexity. The current study also disclosed that around two-thirds of the participants responded that they discontinued medications either sometimes or mostly because of other medicines for

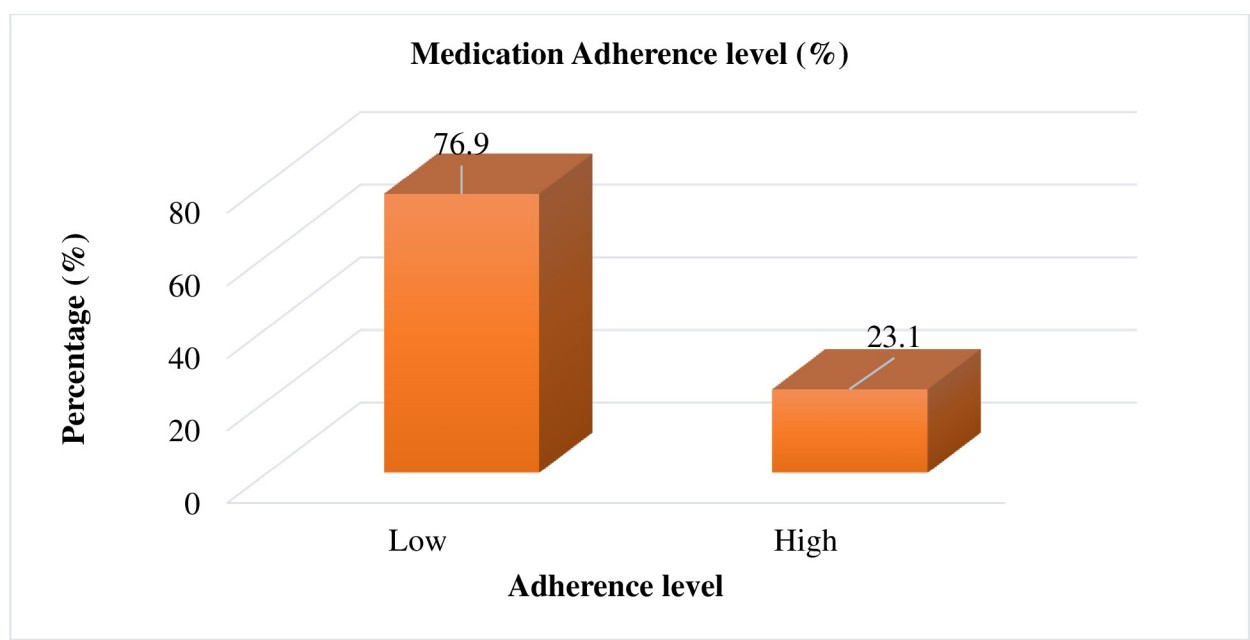

**Fig 1. Medication adherence in T2DM patients with comorbidity at hospitals in Northwest Ethiopia.**

**Table 4. Determinants of medication adherence in patients T2DM patients with comorbidity.**

| Variables | | Adherence level | | 95% CI | | P-value |
|---|---|---|---|---|---|---|
| | | Low | High | COR | AOR | |
| Source of medication cost coverage | Health insurance | 174 | 59 | 2.713(1.433–5.139) | 3.981(0.777–12.036) | 0.003* |
| | Out of pocket | 111 | 11 | 9.284(4.01–21.49) | 10.593(2.682–41.835) | |
| | Free | 25 | 23 | 1 | 1 | |
| Monthly income (ETH.Birr) | < 1500 | 55 | 5 | 11.0(4.057–29.825) | 13.896(2.598–46.199) | <0.001* |
| | 1500–2999 | 46 | 5 | 9.20(3.370–25.116) | 9.369(2.940–25.785) | |
| | 3000–4999 | 160 | 34 | 4.706(2.737–8.092) | 5.095(2.549–13.308) | |
| | ≥ 5000 | 49 | 49 | 1 | 1 | |
| SMBG practice | Yes | 78 | 47 | 0.329(0.203–0.532) | 0.266(0.117–0.604) | 0.002* |
| | No | 232 | 46 | 1 | 1 | |
| Physical activity | Sedentary | 151 | 30 | 1.678(0.893–3.151) | 2.560(0.841–7.794) | 0.053 |
| | Moderate | 96 | 42 | 0.762(0.413–1.406) | 0.809(0.280–2.335) | |
| | Vigorous | 63 | 21 | 1 | 1 | |
| Hypoglycemia | Yes | 30 | 22 | 0.346(0.188–0.636) | 0.662(0.157–2.793) | 0.574 |
| | No | 280 | 71 | 1 | 1 | |
| Antidiabetic medications | Metformin plus glibenclamide plus insulin | 53 | 10 | 0.954(0.358–2.542) | 0.660(0.169–2.581) | 0.410 |
| | Metformin plus insulin | 93 | 38 | 0.441(0.197–0.984) | 0.355(0.120–1.051) | |
| | Metformin pus glibenclamide | 64 | 12 | 0.96(0.375–2.458) | - | |
| | Metformin | 50 | 24 | 0.375(0.159–0.887) | - | |
| | Insulin | 50 | 9 | 1 | 1 | |
| Types of insulin regimens | NPH | 153 | 36 | 2.033 (1.089–3.795) | 1.220(0.453–3.287) | 0.695 |
| | Premixed | 46 | 22 | 1 | 1 | |
| Number of medications | ≤ 3 | 94 | 55 | 0.077(0.023–0.255) | 0.068(0.004–0.813) | 0.014* |
| | 4–5 | 149 | 35 | 0.191(0.057–0.642) | 0.160(0.010–2.520) | |
| | ≥ 6 | 67 | 3 | 1 | 1 | |
| Number of comorbidities | 2 | 127 | 66 | 0.206(0.060–0.703) | 0.307(0.026–0.437) | 0.018* |
| | 3 | 117 | 22 | 0.570(0.159–2.039) | 0.190(0.054–8.927) | |
| | 4 | 38 | 2 | 2.036(0.319–13.006) | 1.514(0.129–8.344) | |
| | ≥ 5 | 28 | 3 | 1 | 1 | |

AOR; Adjusted odds ratio, COR; crude odds ratio, CI; confidence interval

* indicated p value < 0.05.

additional problems and difficulty in remembering because of medication complexity. The other possibilities for their lower medication adherence are the patients' fear of medication adverse effects, medication expensiveness and poor patients' behavior towards their medication. Thus, the finding has implications, which need to be focus towards improving the medication adherence of T2DM patients with comorbidity. In addition, particularly in Ethiopia, the problem might be related to a low level of patients' knowledge about the diabetes, their medications and low socio-economic status, literacy status, cultural and personal perceptions as well as healthcare factors. The previous study also showed that personal beliefs and literacy status have a significant association with medication adherence in patients with chronic illness. Therefore, in Ethiopian settings and population, personal beliefs and literacy status could be addressed in the prescription of medications.

This study showed that the source of cost coverage of medications was significantly associated with levels of medication adherence, which patients who were paid out of pocket for their medication were found more likely to have poor medication adherence compared with

**Table 5. Association of medication adherence and other predicted variables with glycemic control in T2DM patients with comorbidity.**

| Variables | | Glycemic control | | 95% CI | | P-value |
|---|---|---|---|---|---|---|
| | | Poor | Good | COR | AOR | |
| Medication cost coverage | Health insurance | 169 | 64 | 1.886(0.993–3.584) | 0.231(0.021–2.513) | 0.296 |
| | Out of pocket | 104 | 18 | 4.127(1.927–8.836) | 0.694(0.043–11.113) | |
| | Free | 28 | 20 | 1 | 1 | |
| Monthly income (ETH.Birr) | < 1500 | 51 | 9 | 4.617(2.048–10.408) | 1.354(0.114–16.046) | 0.135 |
| | 1501–2999 | 42 | 9 | 3.802(1.670–8.656) | 1.142(0.135–3.308) | |
| | 3000–4999 | 154 | 40 | 3.137(1.849–5.322) | 2.351(0.256–21.616) | |
| | ≥ 5000 | 54 | 44 | 1 | 1 | |
| SMBG practice | Yes | 67 | 58 | 0.217(0.135–0.350) | 0.319(0.056–0.829) | 0.020* |
| | No | 234 | 44 | 1 | 1 | |
| BMI (K/m²) | Low | 27 | 7 | 0.380(0.122–1.186) | 0.435(0.014–1.465) | 0.013* |
| | Normal | 161 | 74 | 0.215(0.094–0.489) | 0.280(0.002–0.474) | |
| | Overweight | 42 | 14 | 0.296(0.111–0.791) | 0.168(0.005–6.249) | |
| | Obese | 71 | 7 | 1 | 1 | |
| Physical activity | Sedentary | 140 | 41 | 0.866(0.459–1.636) | 0.464(0.037–5.891) | 0.823 |
| | Moderate | 94 | 44 | 0.542(0.285–1.030) | 0.501(0.046–5.437) | |
| | Vigorous | 67 | 17 | 1 | 1 | |
| Hypoglycemia | Yes | 29 | 23 | 0.366 (0.201–0.669) | 1.656(0.192–14.310) | 0.647 |
| | No | 272 | 79 | 1 | 1 | |
| Hypertension | Yes | 222 | 65 | 1.600(0.991–2.581) | 0.978(0.143–6.684) | 0.982 |
| | No | 79 | 37 | 1 | 1 | |
| Antidiabetic mediations | Metformin plus glibenclamide plus insulin | 51 | 12 | 0.974(0.393–2.415) | 2.023(0.147–27.786) | 0.878 |
| | Metformin plus insulin | 90 | 41 | 0.503(0.237–1.067) | 0.652(0.071–6.009) | |
| | Metformin pus glibenclamide | 64 | 12 | 1.222(0.497–3.005) | - | |
| | Metformin | 48 | 26 | 0.423(0.188–0.952) | - | |
| | Insulin | 48 | 11 | 1 | 1 | |
| Types of insulin regimens | NPH | 152 | 37 | 2.876 (1.575–5.250) | 3.249(0.534–19.754) | 0.201 |
| | Premixed | 40 | 28 | 1 | 1 | |
| Lipid lowering agents | Atorvastatin | 116 | 27 | 1.953(0.932–4.094) | 4.249(0.706–25.562) | 0.114 |
| | Simvastatin | 33 | 15 | 1 | 1 | |
| Number of medications | ≤ 3 | 95 | 54 | 0.227(0.101–0.510) | 0.076(0.001–4.215) | 0.321 |
| | 4–5 | 144 | 40 | 0.465(0.206–1.050) | 0.227(0.027–1.911) | |
| | ≥ 6 | 62 | 8 | 1 | 1 | |
| Number of medical conditions | 2 | 131 | 62 | 0.616(0.252–1.507) | 5.309(0.384–79.787) | 0.113 |
| | 3 | 112 | 27 | 1.210(0.472–3.100) | 5.858(0.415–82.675) | |
| | 4 | 34 | 6 | 1.653(0.493–5.538) | 1.336(0.134–13.279) | |
| | ≥ 5 | 24 | 7 | 1 | 1 | |
| Level of mediation Adherence | High | 17 | 76 | 0.020(0.011–0.040) | 0.003(0.000–0.113) | 0.002* |
| | Low | 284 | 26 | 1 | 1 | |

AOR; Adjusted odds ratio, COR; crude odds ratio, CI; confidence interval

* indicated p value < 0.05.

patients who received their medication freely. This finding agrees with previous studies [42–44]. The finding indicates that patients who cover medication costs directly form out of pocket may sustain an increase in mediation costs and be forced to withdraw when the medication cost become expensive. In this study, a significant number of participants also responded that

they were discontinued medications because they are not worth for the money and find it difficult to buy medicines because they are expensive. Medication adherence of patients suffers because of high drug costs [45], particularly the problem might be much higher for patients who pay out of pocket. Cost-sharing may deter clinically vulnerable patients from initiating essential medications, compromise adherence and result in treatment failure. Here, patients may benefit from healthcare insurance, which helps them access their medications with optimum pre-paid coverage cost [46, 47]. Therefore, particularly patients with chronic diseases like T2DM may benefit and could be engaged in the Ethiopian health insurance systems with an optimum pre-paid healthcare access coverage cost, which can protect them from catastrophic healthcare expenditures for their medications and treatments. Moreover, this study also showed that patients with low household incomes were found more likely to be low adherent to their medications compared to patients who had relatively high household income. This finding agrees with previous studies [42, 43, 48], which patients with low economic status and household income have the potential to withdraw medications because of affordability issues. This problem is severe in chronic illnesses and patients with comorbidities because of increased medication costs for treating additional conditions. Particularly in Ethiopian settings, most patients are with low socio-economic status [25–28]. In contrary, most patients with T2DM are with comorbid conditions [29]. Thus, this finding indicates that healthcare providers and prescribers could come up with appreciating the socio-economic status of the patients, and clear and good communication towards the affordability of the prescribed medications. The patients may also benefit from the Ethiopian community-based health insurance (CBHI) systems, which may help individuals by providing optimum pre-paid coverage costs and protect them from catastrophic expenditures.

Patients who could practice SMBG were found less likely to have low adherence to their medications compared to patients who did not practice SMBG. This finding implies that patients who practice SMBG can obtain direct feedback on the level of blood glucose and use that information to adjust their choice and help them adhere to their medications. Although the SMBG is an important tool for improving patient self-management and clinicians may use it in guiding medications [49], the current study showed that a significant proportion of patents did not practice SMBG. But the clinical significance of SMBG may depend on the patients understanding of the technical procedures, adherence to the practice, and interpretation of the results. Therefore, patients could be encouraged to practice SBMG, share their testing results with healthcare providers, and the clinicians act towards making treatment decisions [49, 50]. Further, the current study also disclosed that patients with a lower number of medications and medical conditions were found less likely to be low adherent to their medications compared with patients with a higher number of medications and medical conditions. This finding is consistent with previous studies [30, 31, 36, 37], which higher number of medications and medical conditions resulted in low medication adherence because of medication regimen complexity, medication adverse effects, the inability of patients to afford multiple medications. A higher number of medications may also contribute to the loss of the time of administration of medications. Therefore, healthcare providers, in particular prescribers, could focus on practicing with prescribing of optimum number of medications by considering the need of medication treatment of the medical conditions in patients with comorbidity. Patients also need to be highly vigilant and motivated to adhere to their multiple medications, which are necessary to treat the possible and presented comorbidities.

The current study also examined the association of medication adherence and level of glycemic control. In line with previous studies [28, 29, 51–54], majority of patients were under poor glycemic control. Consistent with the previous studies [14, 28, 39, 55], patients who had a low level of medication adherence were found more likely to have poor glycemic control. The

findings may imply that poor glycemic control in the majority of Ethiopian population and settings might be related to low medication adherence. But medication adherence of patients could maximize the effectiveness of pharmaceutical therapy. Thus, patients could be recommended to adhere to their medications. Additionally, patients who could practice SMBG were found less likely to have poor glycemic control compared to patients who didn't practice SMBG. This finding is consistent with previous studies [56–58], which indicate that SMBG can be important in adjusting the level of glycemic control by adhering to medications and taking appropriate measures to improve poor glycemic levels when there are higher blood glucose levels. This finding implies that patients could be recommended to practice SMBG. They also use the SMBG data to adjust their practice, medication adherence and communicate with their healthcare providers and use the data to act on treatment decisions. Moreover, in consistent with previous studies [59–61], patients who had normal BMI were found less likely to have poor glycemic control compared with obese patients. This relation might justify those patients with higher BMI or obesity caused to insulin resistance and in turn, obesity may result in poor glycemic control in the long term. Thus, patients with diabetes could be recommended to reduce their overweight to a normal level by different recommended daily physical activities and modification of diets. In Ethiopia, unhealth sedentary lifestyle has increased and it is among the risk factors of diabetes. Therefore, patients with T2DM could be engaged with an optimum daily physical activity and adjust their diets, and lifestyles.

Generally, this study highlighted the extent of medication adherence and its impact on glycemic control among T2DM patients with comorbidity in resource-limited settings. The findings also have an implication to take measures in the management of T2DM patients with comorbidities. It has explored the medication adherence by assessing patient-behaviors towards their medication adherence, pill/injection burdens due to other medications, and payment related factors to adhere to medications, this tries addressing potential contributors to poor medication adherence in the Ethiopian settings and populations. Indeed, the rapid rise in the prevalence and burden of diabetes mellitus in developing countries, particularly in Ethiopia, where most of the diabetes patients are with comorbid illness and low awareness of the patients towards the disease and medications could seek an urgent intervention towards ensuring medication adherence and achieving glycemic targets. The study may add some background knowledge of the practitioners and patients, and help them towards treatment decisions and modifications accordingly.

The current study has some limitations. The adherence level is determined through patients' self-reported adherence measuring scale, which depends on the honesty and faith in the respondents and could affect the responses resulting in an over or underestimation of the adherence level of medications. Despite this limitation, we hope this study will fill the existing literature gap in the study area and add a body of knowledge to the management practice of T2DM patients with comorbidities.

## Conclusion

The current study concluded that medication adherence was low and significantly associated with glycemic control of patients. Medication cost coverage, monthly income, SMBG practice, number of medications and medical conditions were found to have significant association with medication adherence. On the other hand, glycemic control was found to have a significant association with SMBG practice, level of BMI and level of medication adherence. Therefore, management interventions of T2DM patients with comorbidity should focus on improving medication adherence and other predictor variables.

## Supporting information

**S1 Dataset. Dataset: A data set used to analyze and generate the data.**
(SAV)

## Acknowledgments

The authors want to thank the University of Gondar for providing ethical approval to the study and the selected hospitals for their positive cooperation during the study. We would also like to forward our gratitude to the data collectors and study participants.

## Author Contributions

**Conceptualization:** Ashenafi Kibret Sendekie.

**Data curation:** Ashenafi Kibret Sendekie, Adeladlew Kassie Netere, Asmamaw Emagn Kasahun, Eyayaw Ashete Belachew.

**Formal analysis:** Ashenafi Kibret Sendekie, Adeladlew Kassie Netere, Asmamaw Emagn Kasahun, Eyayaw Ashete Belachew.

**Investigation:** Ashenafi Kibret Sendekie, Eyayaw Ashete Belachew.

**Methodology:** Ashenafi Kibret Sendekie, Adeladlew Kassie Netere, Asmamaw Emagn Kasahun, Eyayaw Ashete Belachew.

**Project administration:** Ashenafi Kibret Sendekie.

**Resources:** Ashenafi Kibret Sendekie.

**Software:** Ashenafi Kibret Sendekie.

**Supervision:** Adeladlew Kassie Netere, Asmamaw Emagn Kasahun, Eyayaw Ashete Belachew.

**Validation:** Adeladlew Kassie Netere, Asmamaw Emagn Kasahun, Eyayaw Ashete Belachew.

**Writing – original draft:** Ashenafi Kibret Sendekie.

**Writing – review & editing:** Ashenafi Kibret Sendekie, Adeladlew Kassie Netere, Asmamaw Emagn Kasahun, Eyayaw Ashete Belachew.

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
