## [Decision Letter · Decision Letter 0]

16 Aug 2022

PONE-D-22-20901Medication adherence and its impact on glycemic control in type 2 diabetes mellitus patients with comorbidity: A multicenter cross-sectional study in Northwest EthiopiaPLOS ONE

Dear Dr. Sendekie,

Thank you for submitting your manuscript to PLOS ONE. After careful consideration, we feel that it has merit but does not fully meet PLOS ONE’s publication criteria as it currently stands. Therefore, we invite you to submit a revised version of the manuscript that addresses the points raised during the review process.

We look forward to receiving your revised manuscript.

Kind regards,

Wanich Suksatan

Academic Editor

PLOS ONE

Journal Requirements:

Reviewers' comments:

Reviewer's Responses to Questions

**Comments to the Author**

1. Is the manuscript technically sound, and do the data support the conclusions?

Reviewer #1: Yes

Reviewer #2: Yes

2. Has the statistical analysis been performed appropriately and rigorously? 

Reviewer #1: Yes

Reviewer #2: Yes

3. Have the authors made all data underlying the findings in their manuscript fully available?

Reviewer #1: Yes

Reviewer #2: Yes

4. Is the manuscript presented in an intelligible fashion and written in standard English?

Reviewer #1: No

Reviewer #2: Yes

5. Review Comments to the Author

Reviewer #1: The overall quality of work by the authors was good. The researchers followed protocol and included detailed information about the process of recruitment, sampling procedures, use of proper measurement tools, data collection and analysis. However, the writing style needs adjustment. Some of the paper sections included long sentences, redundancy, and some grammar and writing errors.

Here are some of my notes on the writing and grammar issues:

-Page 3 / lines 17 to 21: Please rewrite the sentences starting from line 17 and ending in line 21

-Page 3/ Lines 23 to 26: Please rewrite the sentences starting from line 23 and ending in line 26

-Page 3/ Lines 27 to 28: Sentence starting from line 27 and ending in line 28 is not clear, please rewrite

-Page 3/ line 29: word typo, “mediation” = “medication”

-Page 4/ Lines 1 to 5:

Medication adherence in chronic diseases is influenced by patient’s personal and socio-demographic (socio-economic status, age, literacy status, cultural and personal perceptions), healthcare, and facility factors (convenience of pharmacy, medication regimen complexity, clinical characteristics, and number of medical conditions of the patients; 13–17).

-Page 4/ Lines 6 to 7: Please rewrite the sentence.

-Page 4/ Lines 9 to 13: Sentence too long, please rewrite

-Page 4/ Line 17: Please clarify “lower glycemic control”, do you mean “poor or bad glycemic control”

-Page 4/ Line 20 to 23: Please rewrite sentence and separate the sentences at line 21 at “there is no any study ….”

-Page 4/ Lines 24 to 26: Sentences unclear, please rewrite to make it clear.

-Page 5/ Lines 11 to 12: Please rewrite sentence, “The study participants were adults, 18 years or older, diagnosed with T2DM, and volunteered to participate in the study.”

-Page 5/ Lines 27 to 29: Please move sentence to line 24 after sentence ending in line 24 with “… in the setting.”

-Page 6/ Lines 8 to 12: Please rewrite the sentence. Multiple items were repeated in the same sentence.

-Page 11/ Lines 9 to 12: Sentence too long. Please rewrite sentence.

-Page 11/ Lines 12 to 15: Sentence does not make sense. It sounds like low glycemic control can “enhance negative consequences …” Please rewrite sentence to make it clear.

Please be consistent when describing glycemic control: low glycemic control Vs. poor glycemic control.

-Page 11/ Lines 21 to 25: Sentence is repetitive and too long.

Therefore, I believe that the manuscript needs revision for writing. Otherwise, I believe the researchers provided a good and strong study that would add to the diabetes literature because according to their introduction, the literature lacks studies glycemic control and medication adherence information among Ethiopian individuals with type 2 diabetes.

Reviewer #2: Comments to authors

Thank you for an opportunity to review this article. This study aims to identify factors related to medical adherence in patients with type 2 diabetes in Ethiopia. I have some comments as following;

Introduction:

The authors provide the prevalence of diabetes worldwide and in the Ethiopia. However, the introduction should be concise.

Methodology:

Regarding sampling, it is unclear the number of participants from each hospital is unequal. More explanation is required. Did the laboratory test was extracted from medical chart? Who was in charge for data collection?

Regarding the study setting, it might be a good idea to add a short introduction of the study setting to readers at the first part of the methods section,

In outcome measures, it was redundant in explanation of the GMAS instrument. I would rather move GMAS paragraph to outcome measure part. And write; Medication adherence was measured by using GMAS…… to measure how ……….. I think no need for operational definition in the article because it should be included in the instrument explanation. Have the authors asked for permission in using the GMAS instrument?

Results: No comments

Discussion:

The novelty of the findings relative to prior literature should be identified. All factors are common in the medical adherence among diabetes patients and have been published. Discussion should be provided in the context of Ethiopia; why these factors related to medical adherence among Ethiopia population and the study settings.

6. PLOS authors have the option to publish the peer review history of their article (what does this mean?). If published, this will include your full peer review and any attached files.

Reviewer #1: No

Reviewer #2: No

---

## [Author Response · Author response to Decision Letter 0]

22 Aug 2022

Responses to the review’s comments

Dear PLoS ONE Academic Editor,

Thank you for giving us the opportunity to submit a revised draft of the manuscript and we would also like to thank your constructive and fruitful comments and suggestions on our paper (Manuscript ID: PONE-D-22-20901). We are very concerned and combined all the suggested comments provided, which we believe that strengthened the paper and we hope this render our paper to be considered for publication in your reputed journal. We appreciate the time and effort that you and the reviewers dedicated to providing feedback on our manuscript and are grateful for the insightful comments on and valuable improvements to this paper.

We authors would like to let you know that all comments and concerns raised by both academic editors and reviewers are fully addressed and indicated with track changes in the main document and a point-by-point response letter for the editors and reviewers. Moreover, we did our best changes and corrections on this revised manuscript. All the changes and corrections are indicated with track changes in the main document. All page and line numbers refer to the revised manuscript file with tracked changes.

Comments from the editor:

1#.... Journal requirements:

Author reply: Thank you for your recommendations to assure adherence to the manuscript template requirements of the journal. Considering to your recommendation, we have adjusted and ensured it accordingly. All files including the main documents, tables and figures are incorporated according to the journal requirements. 

Response to Reviewers’ comments:

Reviewer #1

#1... The overall quality of work by the authors was good. The researchers followed protocol and included detailed information about the process of recruitment, sampling procedures, use of proper measurement tools, data collection and analysis. However, the writing style needs adjustment. Some of the paper sections included long sentences, redundancy, and some grammar and writing errors.

Author response: We authors are very thankful for your deep concerns and suggestions. We, therefore, accepted the recommendations and made editing and corrections for the whole parts you mentioned. We also go through the whole parts of the manuscript to correct the long sentences, redundancy, and existing grammar and writing errors in the manuscript that indicated with track changes. The manuscript changes have tracked and all the raised issues, changes and responses are indicated with those line numbers in the manuscript with tracked changes. We hope that you find improved. 

Specific comments:

#1.... -Page 3 / lines 17 to 21: Please rewrite the sentences starting from line 17 and ending in line 21

Author reply: Thank you very much and we had revised the sentence, and we hope that you found that it has improved, Page 3, lines 16-19.

#2.... -Page 3/ Lines 23 to 26: Please rewrite the sentences starting from line 23 and ending in line 26

Author reply: Thank you very much and we had revised the sentence. Hopefully, you has found that it has improved, Page 3, lines 21-23.

#3…-Page 3/ Lines 27 to 28: Sentence starting from line 27 and ending in line 28 is not clear, please rewrite

Author reply: Thank you very much and we had revised the sentence, hopefully, you found that it has improved, Page 3, lines 24-25.

#4.... -Page 3/ line 29: word typo, “mediation” = “medication”

Author reply: Thank you very much, and we made it correct, page 3 line 26. 

#5…-Page 4/ Lines 1 to 5: Medication adherence in chronic diseases is influenced by patient’s personal and socio-demographic (socio-economic status, age, literacy status, cultural and personal perceptions), healthcare, and facility factors (convenience of pharmacy, medication regimen complexity, clinical characteristics, and number of medical conditions of the patients; 13–17).

Author reply: Thank you very much for the comment, and we had revised the sentence. Hope you have found that it has improved, Page 3, lines 26-29. 

#6…-Page 4/ Lines 6 to 7: Please rewrite the sentence.

Author reply: Thank you for an important comment, and we had revised the sentence, Page 3, lines 29 and page 4 line 1. 

#7…-Page 4/ Lines 9 to 13: Sentence too long, please rewrite

Author reply: Thank you very much for your valuable comments, and we had revised the sentence, Page 4, lines 3-7.

#8…-Page 4/ Line 17: Please clarify “lower glycemic control”, do you mean “poor or bad glycemic control”

Author reply: We the authors thank you very much for the important comments. We had directly revised it by taking “poor glycemic control”. Thank you in advance again for your valuable comments, Page 4, line 12. 

#9…. -Page 4/ Line 20 to 23: Please rewrite sentence and separate the sentences at line 21 at “there is no any study ….”

Author reply: We are grateful for the comments provided to improve the quality of the paper. Taking your positive comments, we had revised the sentence and separate it. hopefully it has improved, page 4, lines 14-16

#10…. -Page 4/ Lines 24 to 26: Sentences unclear, please rewrite to make it clear.

Author reply: Thank you for making the sentence clear and rewrite it. The comments improve the quality of the paper. Taking your positive comments, we had corrected the sentence. And hopefully it has improved and clear, Page 4, lines 18-21. 

#11…. -Page 5/ Lines 11 to 12: Please rewrite sentence, “The study participants were adults, 18 years or older, diagnosed with T2DM, and volunteered to participate in the study.”

Author reply: Thank you very much for the comments to improve the quality of the paper. Taking your positive comments, we had rewritten the sentence. And hopefully it has improved and clear, Page 5, line 6.

#12…. -Page 5/ Lines 27 to 29: Please move sentence to line 24 after sentence ending in line 24 with “… in the setting.”

Author reply: Thank you for an interesting comment, which make it coherent and readable. Considering your valuable comment, we had revised by moving it to the provided space, Page 5, lines 20-22.

#13…-Page 6/ Lines 8 to 12: Please rewrite the sentence. Multiple items were repeated in the same sentence.

Author reply: Thank you very much for your comments to rewrite and correct the repeated items in the sentence. By considering your comments and suggestion, we had revised and rewrite the sentence. Hope you found that it has improved, Page 6, lines 4-8.

#14…. -Page 11/ Lines 9 to 12: Sentence too long. Please rewrite sentence.

Author reply: Thank you for such valuable comment, and we had revised and rewrite the sentence, Page 11, lines 2-6.

#15…-Page 11/ Lines 12 to 15: Sentence does not make sense. It sounds like low glycemic control can “enhance negative consequences …” Please rewrite sentence to make it clear.

Author reply: Thank you very much and taking your positive comments, we wrote the sentence. And hopefully it has improved and clear, Page 11, lines 6-8.

#16…-Please be consistent when describing glycemic control: low glycemic control Vs. poor glycemic control.

Author reply: Thank you for such an important comment. By considering your comments suggestion, we had revised and described glycemic control with poor glycemic control.

#17…-Page 11/ Lines 21 to 25: Sentence is repetitive and too long. 

Author reply: Thank you very much for your comments and suggestions to rewrite and correct the repeated words in the sentence and summarized the long sentence. By considering your comments and suggestion, we had revised and summarized the long sentence, Page 11, lines 12-16.

Reviewer #2

Comments to authors

Thank you for an opportunity to review this article. This study aims to identify factors related to medical adherence in patients with type 2 diabetes in Ethiopia. I have some comments as following;

Introduction:

#1… The authors provide the prevalence of diabetes worldwide and in the Ethiopia. However, the introduction should be concise.

Author reply: We authors are very grateful for the concerns and recommendations you raised to concise the introduction. Considering your valuable comments and suggestions, we revised the introduction section and tried to concise. The changes are found with tracks in the manuscript with track changes. We hope, you have found that it has improved. 

Methodology:

#2… Regarding sampling, it is unclear the number of participants from each hospital is unequal. More explanation is required. 

Author reply: Thank you for your comments and concerns regarding the number of the participants involved from each hospital. As it was mentioned in the section, participants from each hospital were recruited in the study based on the number of patients with T2DM. It was just to take a representative sample from each hospital. The number of patients with T2DM in each hospital was taken from records of the previous three follow-up months in the settings. Taking your constructive comments and suggestions, we have revised this section and tried to cleat it, page 5, lines 17-22. Hopefully, you have found that it has improved.

#3… Did the laboratory test was extracted from medical chart? 

Author reply: Thank you very much to your comments to clear the data collection process regarding laboratory tests. As we sated in the data collection sections, secondary data like laboratory test records were extracted from medical records of the patients. Yes, as you said laboratory test of the participants was extracted from medical records of the patients. In all of the selected hospitals, medical records of the participants are available in physical medical chart. Therefore, laboratory test results of the participants were extracted from printed laboratory results in the medical chart of the patients. 

#4…. Who was in charge for data collection?

Author reply: Thank you for your comments and questions regarding data collectors. As we tried to mention in the data collection producers and instruments’ section, the data was collected by pharmacists and nurses from the selected hospitals after they had received training regarding to the purpose of the study, data collection procedures, the nature of the data collection instruments and ethical aspects of the study. They had involved as a data collector in voluntarily. Considering your questions and concerns, we have revised and clear the statements, page 6, lines 10-12. 

#5…Regarding the study setting, it might be a good idea to add a short introduction of the study setting to readers at the first part of the methods section,

Author reply: Thank you for your important information which can add some basic information regarding the study area to the readers. Bay taking your valuable suggestions, we have revised the section and included some basic background information, page 4, lines 25-29 & page 5 lines 1-5.

#6… In outcome measures, it was redundant in explanation of the GMAS instrument. I would rather move GMAS paragraph to outcome measure part. And write; Medication adherence was measured by using GMAS…… to measure how ……….. I think no need for operational definition in the article because it should be included in the instrument explanation. 

Author reply: Thank you to your suggestion to avoid the redundancy. We have totally accepted your important suggestions and revised accordingly. We had also avoided the operational definitions and included in the instrument explanation. The changes are presented in the tracked manuscript. 

#7…Have the authors asked for permission in using the GMAS instrument?

Author reply: Thank you for your positive and important concerns and suggestions. We can ask and access the copyright permission of the GMAS instrument if the paper will be accepted for publication. We hope, we can do it and will happen if the paper will be accepted for publication. Thank you in advance for your positive comments and concerns. 

Discussion 

#8… The novelty of the findings relative to prior literature should be identified. All factors are common in the medical adherence among diabetes patients and have been published. Discussion should be provided in the context of Ethiopia; why these factors related to medical adherence among Ethiopia population and the study settings.

Author reply: We the authors are grateful for the comments provided to improve the quality of the paper. Taking your positive comments, we have tried to incorporated novelty of the findings and revised the discussion on the context of Ethiopian population and settings. Hopefully, you have found that it has improved. All the changes are presented in the tracked manuscript.

---

## [Decision Letter · Decision Letter 1]

6 Sep 2022

PONE-D-22-20901R1Medication adherence and its impact on glycemic control in type 2 diabetes mellitus patients with comorbidity: A multicenter cross-sectional study in Northwest Ethiopia

PLOS ONE

Dear Dr. Sendekie,

Thank you for submitting your manuscript to PLOS ONE. After careful consideration, we feel that it has merit but does not fully meet PLOS ONE’s publication criteria as it currently stands. Therefore, we invite you to submit a revised version of the manuscript that addresses the points raised during the review process.

We look forward to receiving your revised manuscript.

Kind regards,

Wanich Suksatan

Academic Editor

PLOS ONE

Journal Requirements:

Additional Editor Comments:

Based on the reviewer's opinions, I agreed with the reviewer 1 that some minor issues need to be revised before accepted to publish.

Reviewers' comments:

Reviewer's Responses to Questions

**Comments to the Author**

1. If the authors have adequately addressed your comments raised in a previous round of review and you feel that this manuscript is now acceptable for publication, you may indicate that here to bypass the “Comments to the Author” section, enter your conflict of interest statement in the “Confidential to Editor” section, and submit your "Accept" recommendation.

Reviewer #1: All comments have been addressed

Reviewer #2: All comments have been addressed

2. Is the manuscript technically sound, and do the data support the conclusions?

Reviewer #1: Yes

Reviewer #2: Yes

3. Has the statistical analysis been performed appropriately and rigorously? 

Reviewer #1: Yes

Reviewer #2: Yes

4. Have the authors made all data underlying the findings in their manuscript fully available?

Reviewer #1: Yes

Reviewer #2: Yes

5. Is the manuscript presented in an intelligible fashion and written in standard English?

Reviewer #1: Yes

Reviewer #2: Yes

6. Review Comments to the Author

Reviewer #1: I would like to thank the authors for their consideration of my comments and working on it accordingly. However, I would suggest that some of the parts detected in the comments were not properly addressed. Specifically, I would like from the authors to address the following suggestions:

Page 3/ Lines 26-29: The sentence included the word "patients" three times which is not needed. Also, the end of the sentence to write "are factors that influence medication adherence ..."

Page 3/ Line 29, Page 4/ lines 1-3: I would suggest choosing the single percentage or the range of percentages if I am allowed to.

Page 5/ Lines 6-8: You could start the sentence with: to be included in the study, participants should be adults (aged 18 years or older), diagnosed with diabetes, and are diagnosed with at least one comorbidity.

Page 4/ line 6, Page 11/ Line 7: the sentence "enhance negative consequences" does not make sense. From my point of view, you cannot enhance negative things. Therefore, I would suggest changing to possibly "increase the risk for negative consequences."

I hope that my review is of assistance to you. Thank you for your hard work and I hope everything works good for your team.

Please accept my best regards,

Reviewer #2: Authors has rewritten all issues according to suggestion. This study is very interesting and will add some valuable knowledge in the diabetes study.

7. PLOS authors have the option to publish the peer review history of their article (what does this mean?). If published, this will include your full peer review and any attached files.

Reviewer #1: No

Reviewer #2: No

---

## [Author Response · Author response to Decision Letter 1]

7 Sep 2022

Responses to the review’s comments

Dear PlOS ONE Academic Editor,

Thank you for giving us the opportunity to submit a revised draft of the manuscript and we would also like to thank your constructive and fruitful comments and suggestions on our paper (Manuscript ID: PONE-D-22-20901R1). We are very concerned and combined all the suggested comments provided, which we believe that strengthened the paper and we hope this render our paper to be considered for publication in your reputed journal. We appreciate the time and effort that you and the reviewers dedicated to providing feedback on our manuscript and are grateful for the insightful comments on and valuable improvements to this paper.

We authors would like to let you know that all comments and concerns raised by both academic editors and reviewers are fully addressed and indicated with track changes in the main document and a point-by-point response letter for the editors and reviewers. 

Comments from the editor:

1#.... Journal requirements:

Author reply: Thank you for your comments and recommendations to ensure that the reference is complete and correct. 

We have checked the whole references using Google scholar searching. Consequently, we have found some errors on reference 13. Accordingly, we had made it correct and we also include the DOI and indicate the change with track changes. In addition, when we have searched reference number 32 on Google scholar, we have not found it and then we have looked the original article on google searching. Now we ensured it was correctly cited. We also include the DOI. Other than these two references, we have not found any error on our best searching, all listed references are found on Google scholar. However, in case any retracted reference is still included, it is not doing purposely and it might be unknowingly. If so, it would be my pleasure if you indicate me which one is retracted. Thank you in advance for your hard work and positive comments. 

Additional Editor Comments:

2#.......Based on the reviewer's opinions, I agreed with the reviewer 1 that some minor issues need to be revised before accepted to publish.

Author response: We authors are very thankful for your concerns and suggestions. We had revised the raised issues and the changes are tracked. The manuscript changes have tracked and all the raised issues, changes and responses are indicated with those line numbers in the manuscript with tracked changes. We hope that you have found it has improved.

Response to Reviewers’ comments:

Reviewer #1

I would like to thank the authors for their consideration of my comments and working on it accordingly. However, I would suggest that some of the parts detected in the comments were not properly addressed. Specifically, I would like from the authors to address the following suggestions:

#1... Page 3/ Lines 26-29: The sentence included the word "patients" three times which is not needed. Also, the end of the sentence to write "are factors that influence medication adherence ..."

Author response: We authors are very thankful for your deep concerns and suggestions. We have accepted the recommendations and made editing and corrections for the whole parts you mentioned. The manuscript changes have tracked and all the raised issues, changes and responses are indicated with those line numbers in the manuscript with tracked changes. We hope that you have found that it has improved. Page 3/ Lines 26-29.

#2.... Page 3/ Line 29, Page 4/ lines 1-3: I would suggest choosing the single percentage or the range of percentages if I am allowed to.

Author reply: Thank you very much. It was to implicate rate of medication nonadherence to patients with comorbidity and patients with T2DM, respectively. But based on your recommendation and it was already mentioned earlier, we had revised and choose the one with ranges of percentages. Page 3/lines 29 and page 4/lines 1-2.

#3…Page 5/ Lines 6-8: You could start the sentence with: to be included in the study, participants should be adults (aged 18 years or older), diagnosed with diabetes, and are diagnosed with at least one comorbidity.

Author reply: Thank you very much for your positive comments for improving the quality of paper. Base on your kind recommendation, we had revised the sentence. Page 5/lines 5-6.

#4.... Page 4/ line 6, Page 11/ Line 7: the sentence "enhance negative consequences" does not make sense. From my point of view, you cannot enhance negative things. Therefore, I would suggest changing to possibly "increase the risk for negative consequences."

Author reply: Thank you for your positive comments. We made it correct and revised accordingly, page 4/line 4 and page 11/line 5.

Generally, we the authors are grateful for the comments provided to improve the quality of the paper. We also have a great appreciation for your efforts and assistance for improving the paper. Thank you in advance. 

Reviewer #2

Authors has rewritten all issues according to suggestion. This study is very interesting and will add some valuable knowledge in the diabetes study.

Author reply: We the authors are grateful for the comments provided to improve the quality of the paper. We also have a great appreciation for your efforts and assistance for improving the paper. Thank you in advance.

---

## [Editor Report · Decision Letter 2]

8 Sep 2022

Medication adherence and its impact on glycemic control in type 2 diabetes mellitus patients with comorbidity: A multicenter cross-sectional study in Northwest Ethiopia

PONE-D-22-20901R2

Dear Dr. Sendekie,

We’re pleased to inform you that your manuscript has been judged scientifically suitable for publication and will be formally accepted for publication once it meets all outstanding technical requirements.

Kind regards,

Wanich Suksatan

Academic Editor

PLOS ONE

Additional Editor Comments (optional):

Congratulations for successful amendments!

---

## [Editor Report · Acceptance letter]

12 Sep 2022

PONE-D-22-20901R2 

Medication adherence and its impact on glycemic control in type 2 diabetes mellitus patients with comorbidity: A multicenter cross-sectional study in Northwest Ethiopia 

Dear Dr. Sendekie:

I'm pleased to inform you that your manuscript has been deemed suitable for publication in PLOS ONE. Congratulations! Your manuscript is now with our production department. 

Kind regards, 

on behalf of

Dr. Wanich Suksatan 

Academic Editor

PLOS ONE